# Integrated multi-omic analyses provide insight into colon adenoma susceptibility modulation by the gut microbiota

Susheel Bhanu Busi,[1,2] Zhentian Lei,[3,4] Lloyd W. Sumner,[3,4] James M. Amos-Landgraf[1,5,6]

**ABSTRACT** Colon cancer onset is strongly associated with the differences in microbial taxa in the gastrointestinal tract. Although recent studies highlight the role of individual taxa, the effect of a complex gut microbiome (GM) on the metabolome and host transcriptome is still unknown. We used a multi-omics approach to determine how differences in the GM affect the susceptibility to adenoma development in a rat model of human colon cancer. Ultra-high performance liquid chromatography mass spectrometry of feces collected prior to observable disease onset identified putative metabolite profiles that likely predict future disease severity. Transcriptome analyses performed after disease onset from normal colonic epithelium and tumor tissues show a correlation between GM and host gene expression. Integrated pathway analyses of the metabolome and transcriptome based on putatively identified metabolic features indicate that bile acid biosynthesis is enriched in rats with high tumors along with increased fatty acid metabolism and mucin biosynthesis. Targeted pyrosequencing of the Pirc allele indicates that the GM alters the mechanism of adenoma development and may drive an epigenetic pathway of tumor suppressor silencing. This study reveals how untargeted metabolomics identifies signatures of susceptibility and integrated analyses uncover pathways of differential mechanisms of loss of tumor suppressor gene function and for potential prevention and therapeutic intervention.

**IMPORTANCE** The association between the gut microbiome and colon cancer is significant but difficult to test in model systems. This study highlights the association of differences in the pathogen-free gut microbiome to changes in the host transcriptome and metabolome that correlate with colon adenoma initiation and development in a rat genetic model of early colon cancer. The utilization of a multi-omics approach integrating metabolomics and transcriptomics reveals differences in pathways including bile acid biosynthesis and fatty acid metabolism. The study also shows that differences in gut microbiomes significantly alter the mechanism of adenoma formation, shifting from genetic changes to epigenetic changes that initiate the early loss of tumor suppressor function. These findings enhance our understanding of the gut microbiome's role in colon cancer susceptibility, offer insights into potential biomarkers and therapeutic targets, and may pave the way for future prevention and intervention strategies.

**KEYWORDS** microbiome, colon cancer, transcriptomics, metabolomics, integrate analyses, pyrosequencing, loss of heterozygosity

Colorectal cancer is the second leading cause of cancer death and remains difficult to diagnose without invasive or non-universally available procedures such as colonoscopy (1). Several recent studies in animal models and human patient populations have begun to identify biomarkers that have some diagnostic capability (2–5). Additionally, association studies in patient populations have shown positive and negative correlations with various bacterial species (6, 7). The association between

Address correspondence to Susheel Bhanu Busi, susheel.busi@uni.lu, or James M. Amos-Landgraf, amoslandgrafj@missouri.edu.

The authors declare no conflict of interest.

See the funding table on p. 17.

certain bacterial species and a quantifiable impact on colon adenoma development has also been shown in animal models (8, 9). However, the link between early diagnostic biomarkers and the gut microbiota has not been sufficiently investigated and the mechanisms driving phenotypic differences are not well determined. The diagnostic ability for biomarkers to correlate with early colon cancer is likely owing, at least in part, to bacterially derived metabolites and the corresponding host responses to these metabolites (10, 11).

Untargeted metabolomics is a maturing field focused on the large-scale quantitative and qualitative analyses of small molecular weight (<2,000) biomolecules (12). Information from these studies provides unique insight into physiological pathways that have important roles in health and disease (13). Given that microbial species play a critical role in both production and use of host metabolites, it is likely that the GM has a substantial impact on the overall metabolite composition (14). Confirming this hypothesis, studies have demonstrated significant differences in metabolites between germ-free mice and their conventionally housed counterparts, emphasizing a microbiota-driven metabolic profile (15). As a result, the role of metabolic mediators as intermediates between the GM and tumorigenesis in both rodent models and humans has garnered substantial interest. Dazard et al. used mass spectrometry to determine that plasma from $Apc^{Min}$ mice had a distinct metabolome compared to wild-type (WT) littermates (16). However, due to a lack of longitudinal metabolomics data in this study and others, it is unclear whether these metabolic changes are a consequence of tumor development or are causative of tumor initiation or progression.

We previously reported that naturally occurring gut microbiota (GM) modulated colon cancer susceptibility in a preclinical rat model of familial adenomatous polyposis (9). We rederived isogenic embryos of the F344/NTac-Apc$^{+/Pirc}$ rat model into different populations of surrogate dams each harboring distinct gut microbiota: GM:F344, GM:SD, and GM:LEW. Through this method, we created animals that harbored distinct endogenous complex GMs. Pirc rats with the GM:F344 had the highest tumor burden, while GM:LEW rats had a significantly reduced tumor burden, including two animals that had no visible colonic adenomas at 6 months of age (9). Previous studies examining the mechanism of loss of the wild-type Apc allele in Pirc animals revealed that multiple pathways of adenoma development exist, but it is unknown how the GM could potentially influence this process (17). The GM and metabolome separately have been shown to affect colon cancer tumorigenesis; however, there are insufficient data demonstrating how host gene expression is affected by the interplay of the microbiome and intestinal metabolites. We used a multi-omics approach to evaluate how differences in the microbiome affect the fecal metabolome and host gene expression to identify putative indicators of risk for adenoma development and begin to reveal the mechanisms by which the GM modulates disease susceptibility.

## RESULTS

### Quantitative pyrosequencing identifies differences in *Apc* loss of function

To determine if the GM altered how the wild-type *Apc* function was lost, we performed quantitative targeted pyrosequencing of the Pirc mutation in DNA and RNA samples extracted from tumors from GM:F344 and GM:LEW adenomas and adjacent normal epithelium since these two groups presented the highest and lowest number of tumors. Adenomas from the GM:LEW group exhibited loss of heterozygosity most often (20/21), whereas nearly 30% of adenomas from the GM:F344 group maintained heterozygosity at the *Apc* locus (7/24) (Fig. 1A). We further performed allele-targeted pyrosequencing using cDNA from RNA extracted from tumors that maintained the wild-type allele to determine if both alleles were expressed or if only the mutant allele was expressed. The majority of adenomas that maintained heterozygosity exhibited monoallelic expression of the mutant *Apc* allele while maintaining the WT allele (5/7) (Fig. 1B).

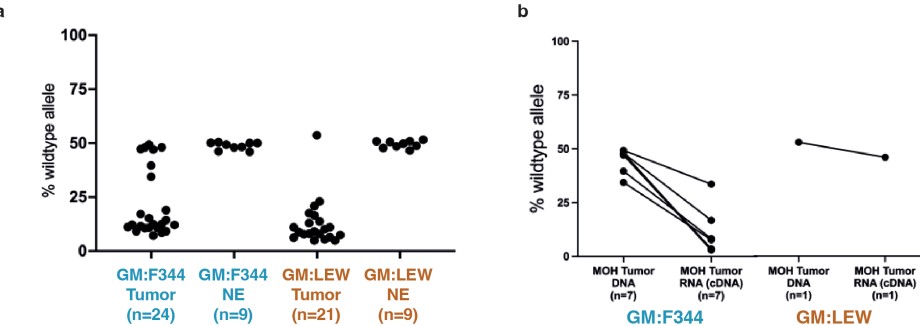

**FIG 1** Quantitative pyrosequencing that differentiates the mutant and Pirc allele. (A) DNA isolated from tumor or adjacent normal tissues from GM:F344 or GM:LEW indicates that the majority of tumors undergo loss of heterozygosity (LOH) in the adenomas while both alleles are maintained in adjacent normal epithelium. Tumors that have greater than 30% presence of the WT allele are designated as maintaining heterozygosity (MOH). (B) Comparison of the allele ratios of the DNA from tumors classified as MOH with the relative expression of the two alleles from RNA isolated from tumors.

## Metabolite features at 1 month of age are correlated with tumor susceptibility and severity at later developmental stages

Fecal samples collected from rederived Pirc rats harboring distinct GMs at 1 month of age were analyzed by ultra-high performance liquid chromatography and mass spectrometry (UHPLC-MS) and found an average of 499 raw features. Of these, 246 putative metabolites were shared among all samples and were used for subsequent analyses (Table S1). Principal component analysis (PCA) indicated a separation of 33.2% along the first component (Dimension 1) accounting for some of the variabilities between each group (Fig. 2A). Furthermore, a hierarchical clustering analysis using a combination of Euclidean distance and Ward's clustering algorithm on the metabolomics data set was used to identify the (dis)similarity of the samples within the respective groups and to each other. Interestingly, the clustering demonstrated a separation of the metabolomic profile of the fecal samples at 1 month which correlated with the number of colonic adenomas that the rats would eventually develop at 6 months of age (Fig. 2B). Here, we found that two major groups of samples were observed, where the clustering was split based on Pirc rats that had ≤9 adenomas at 6 months versus those that developed ≥19 adenomas on average.

## Metabolomics analyses indicate differential metabolic profiles between GM:F344 and GM:LEW

Based on the observation that GM:F344 and GM:LEW had the highest and lowest average number of tumors, respectively, we further analyzed the differential features contributing to disease susceptibility only within these groups. A heatmap leveraging clustering analyses was used to classify the metabolic profiles of GM:F344 and GM:LEW, revealing two independent metabolic profiles (Fig. 3A). Several metabolites were also differentially abundant in each of the groups (Fig. 3B). Linear discriminant analyses (LDAs) identified the putative metabolites contributing to the high (GM:F344) and low (GM:LEW) tumor groups' separation observed in the dendrogram at 1 month of age (Fig. 3C). Some of the putative metabolites identified in the low tumor group, i.e., GM:LEW, showed up to a fourfold increase compared to GM:F344 high tumor group (Fig. 3D). To further identify the compounds that were differential between the low and high tumor groups, tandem MS spectra were generated for the compounds with the mz/rt values of 329.10/9.2 min and 315.12/6.39 min. However, their identities could not be definitively established based on the spectral libraries currently available. Interestingly, comparison of the features without regard to specific GM and terminal tumor counts found significant correlations between individual metabolites at 1 month of age and terminal colonic tumor numbers (Fig. 3E).

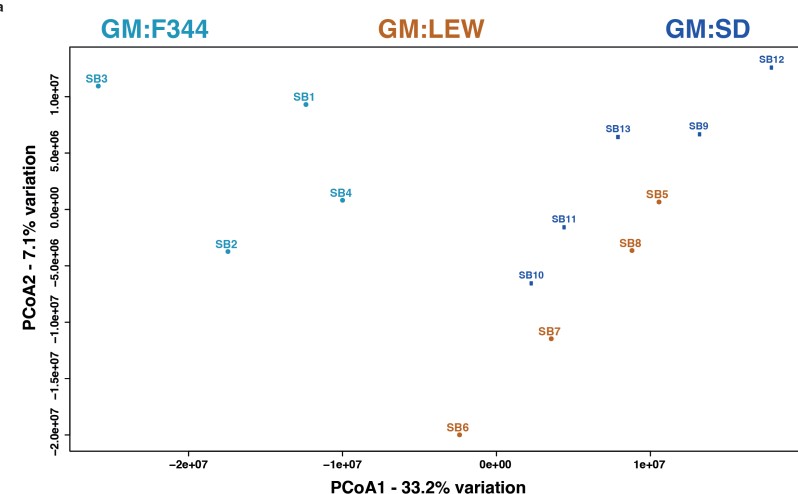

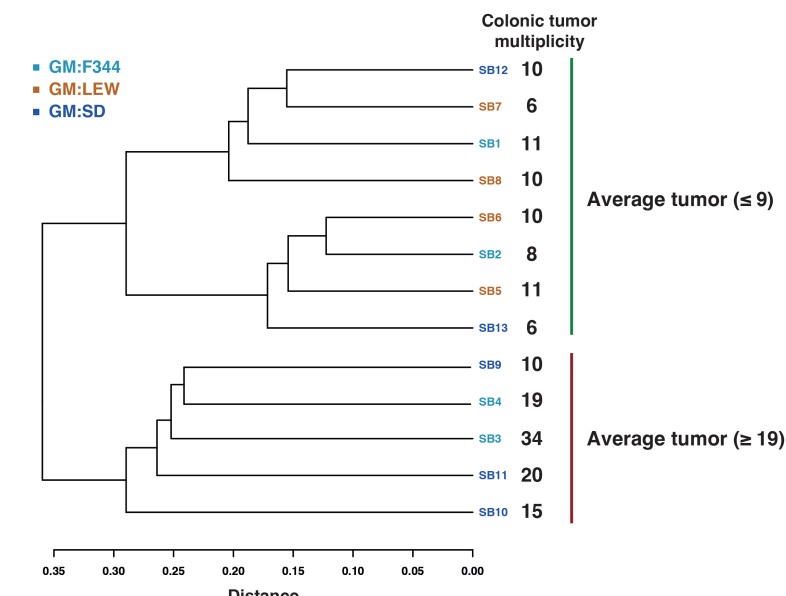

**FIG 2** Metabolomics analyses indicate differential features in feces from differing gut microbiota. (A) Principal coordinate analysis (PCoA) depicting the three groups, viz., GM:F344 (cyan), GM:SD (blue), and GM:LEW (brown), demonstrates that the samples cluster independent of either group. PCoA1 and PCoA2 refer to the amount of variation observed between the groups, with the respective percentage of variation explained. (B) Dendrogram analysis was performed on the putative metabolite features using the Euclidean distance of measurement and Ward's clustering algorithm. The major root of the tree separated two samples from the remaining six, irrespective of either GM profile. Retrospectively, it was established that the clustering analysis was based on the colonic tumor multiplicity, indicated by the numbers adjacent to the dendrogram. The two clusters separated based on animals with an average of nine tumors or those with greater than 19 colonic tumors on average.

## Bile acid biosynthesis and aspirin-triggered resolvin E biosynthesis pathways are associated with putative fecal metabolomics features

Putative identifications for the differential metabolite features listed in Table 1 are based on the METLIN metabolite library available for public access at the time of the analyses. Based on relative mass defect (RMD) values, four putative metabolites were classified as steroids while the others were classified as polyphenols, carbohydrates, short-chain fatty acids, and flavonoids. All putative features identified using UHPLC-MS were subjected to

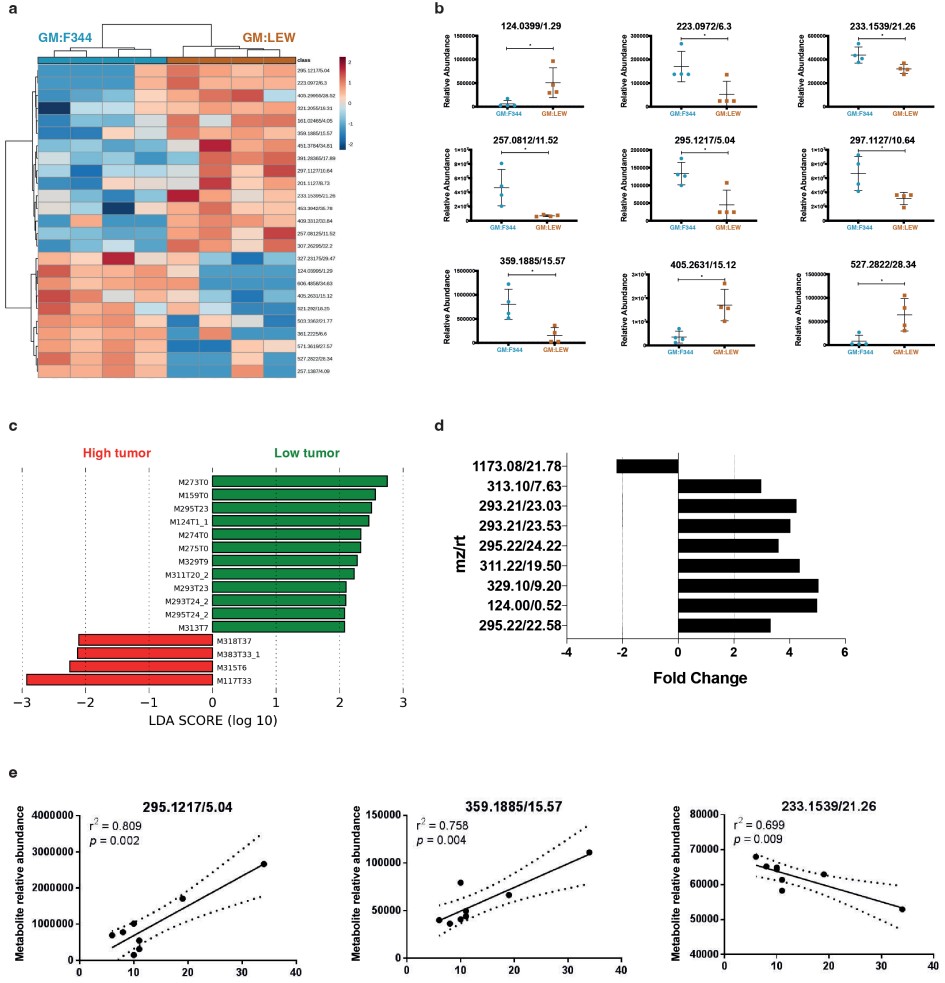

**FIG 3** Metabolite features at 1 month of age may predict tumor susceptibility and severity. (A) Metabolite features that were significantly different between the high (GM:F344) and low tumor (GM:LEW) groups were used to generate a heatmap illustrated with the samples along the x-axis and the metabolite features along the y-axis. Hierarchical clustering was performed based on samples and indicates that the GM:F344 samples cluster separately from the GM:LEW group. The fold-change is represented by intensity with red being an increased fold change while blue refers to a decrease. (B) The relative abundance of the top nine differential metabolites between GM:F344 (cyan dots) and GM:LEW (brown squares) is depicted. (C) LDA and (D) fold-change analysis were used to identify the metabolites driving the dendrogram tree separation in (A) and differential modulation in high and low tumor groups. (E) Correlation analysis was performed using Pearson's method to determine positively and negatively correlating metabolites that are associated with increased or decreased tumor multiplicity.

pathway analysis to identify KEGG pathways that were significantly modulated between the two GM profiles. Bile acid biosynthesis (neutral pathway) and aspirin-triggered resolvin E biosynthesis were most differentially affected (Fig. 4). The pathway analyses also identified potential genes that may affect or may be affected by these putative metabolites (Table 2). The putative identities for the metabolites affecting the bile acid and resolvin E biosynthesis pathways include secondary bile acids (SBAs) such as glycocholate, glycochenodeoxycholate, and 7α-hydroxycholest-4-en-3-one (Table 3). We sampled an independent population of Pirc and WT rats at 1 month of age to validate the bile acid and resolvin E biosynthesis pathways as being risk factors for eventual development of adenomas and to determine if these can be observed in serum. We found that the Pirc animals had a relatively different metabolomic profile than the WT

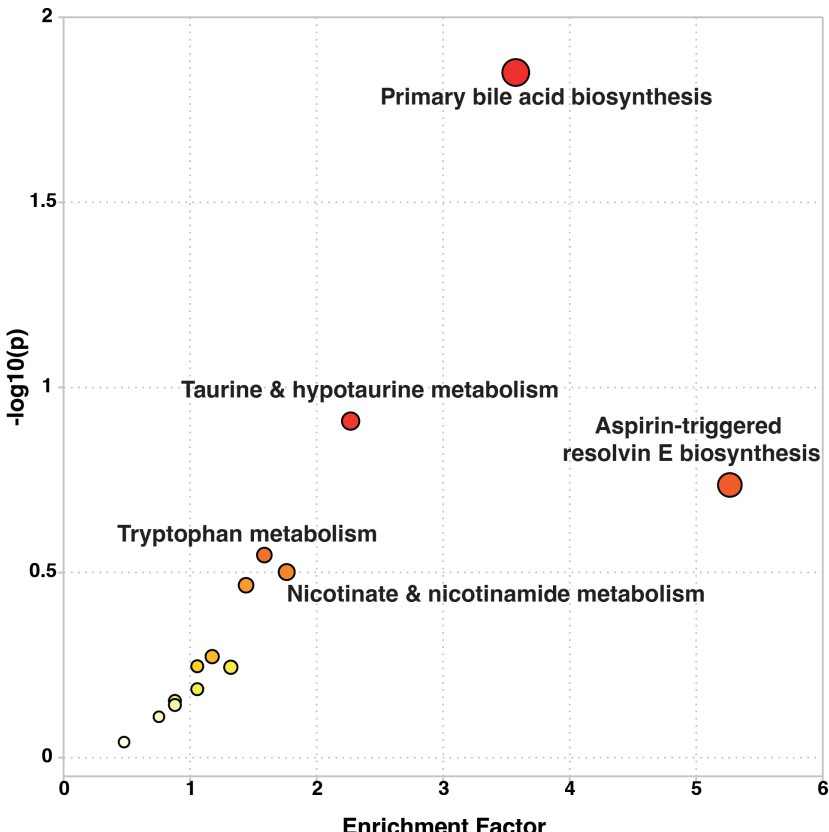

| Pathway | Genes | Enzyme activity | Group |
|---|---|---|---|
| Bile acid biosynthesis (neutral pathway) | CYP8B1 | 7-alpha-hydroxycholest-4-en-3-one 12-beta-hydroxylase | GM:LEW |
| | BAAT | bile acid-CoA: amino acid N-acyltransferase | |
| Aspirin-triggered resolving E biosynthesis | PTGS2 | 18R-hydro(peroxy)-EPE synthase | GM:F344 |
| | ALOX5 | 5S hydroperoxy HEPE synthase | |

**FIG 4** Bile acid biosynthesis and aspirin-triggered resolvin E biosynthesis pathways are most affected by metabolite features. Systems biology analyses, taking into account the differential putatively identified metabolites. The results showed that bile acid biosynthesis (neutral pathway) and the aspirin-triggered resolvin E biosynthesis were significantly different ($P < 0.01$, Student's $t$-test) between GM:F344 and GM:LEW. The $P$-value is indicated along the y-axis. Table in the figure lists the genes and the respective enzymes demonstrating significant differences and the groups in which the pathways are enriched are indicated.

rats (Fig. S1A), and showed elevated serum levels of metabolites related to the bile acid pathway (Fig. S1B).

## Gut microbiota is correlated with differences in gene expression in both the normal colonic epithelium and tumor tissues

To understand the effect of the GM and also the respective metabolites on the host transcriptome, RNA sequencing analyses (RNASeq) were performed on normal colonic epithelium (NE) and tumor (T) tissues after sacrifice at 6 months of age. Expectedly, the NE transcriptome profiles in the GM:F344 and GM:LEW Pirc rats were similar to each other, while the tumor tissues in both groups expressed similar genes (Fig. 5A). A clear separation, in this context, was observed between the respective NE and T

**TABLE 1** Compound class, RMD, and putative identification of metabolites features in the METLIN databases[a]

| Mass-charge/retention time (mz/rt) | Chemical formula | Relative mass defect (RMD) | Compound class | Putative identification (METLIN ID) |
|---|---|---|---|---|
| 124.03995/1.29 | C6H9O3 | 322.0737 | Polyphenol | NA[b] |
| 223.0972/6.3 | C7H3N2O7 | 435.6845 | Carbohydrate | NA |
| 233.15395/21.26 | C15H22O2 | 660.2933 | Steroid | 90173 |
| 257.08125/11.52 | C17H8NO2 | 316.0479 | Polyphenol | NA |
| 295.1217/5.04 | C18H17NO3 | 412.3723 | Carbohydrate | 95663 |
| 297.1127/10.64 | C18H18O4 | 379.3173 | Flavonoid | 52682 |
| 359.1885/15.57 | C24H25NO2 | 524.7941 | Steroid/SCFA[c] | 675713 |
| 405.2631/15.12 | C24H36O5 | 649.2079 | Steroid | 84737 |
| 527.2822/28.34 | C13N13O12 | 535.1973 | Steroid/SCFA[c] | NA |

[a]LC-MS analysis between groups identified several putative metabolites that are listed in the table as mass-charge to retention time ratios. Chemical formulas generated through the Bruker software, along with the calculated RMD and compound classes, are also identified. This is additionally supplemented with the putative identification based on the METLIN library.
[b]NA, not available.
[c]SCFA, short-chain fatty acid.

tissues coupled with several differentially expressed genes (Fig. 5B; Tables S2 and S3). For example, 4,809 genes were differentially expressed between NE and T within the GM:F344 group, while 6,322 genes were differentially expressed between NE and T within GM:LEW. Importantly, we found that between the GM:F344 and GM:LEW NE samples, there were 2,046 differentially expressed genes, while 2,752 genes were found to be significantly differentially expressed between tumors from GM:F344 compared to that of GM:LEW.

## Pathway analyses identify potential mechanisms contributing to high and low colonic tumor susceptibility

Pathway analysis using the differentially expressed genes found an enrichment in many cancer- and metabolism-related pathways, including the fatty acid and the mucin type-O glycan biosynthesis pathways. An increased pathway topology in the normal epithelium from the high tumor, i.e., GM:F344 group, (Fig. S2A) was observed, suggesting a significant role and position of the genes in the pathway. Increased cell cycle, RNA transport, and TCA cycle pathways were also observed in the normal epithelium of GM:F344. Conversely, normal epithelium from the GM:LEW (low tumor) profile showed an increase in apoptotic pathways along with fat digestion and absorption and calcium signaling pathways (Fig. S2B).

To integrate the metabolomics and transcriptome profiles, we determined the expression differences of the genes contributing to the predicted putative metabolic pathways, i.e., bile acid biosynthesis and aspirin-triggered resolvin E biosynthesis (Fig. 4; Table 2; Fig. S3A). We examined the gene expression involved in the resolvin E biosynthesis pathway and found that *Ptgs2* was significantly increased in the normal epithelium tissues of the high tumor group (GM:F344) compared to the GM:LEW group (Fig. 5C). Interestingly, *Ptgs2* was highly elevated in tumor tissues of the low tumor group (GM:LEW) at 6 months of age (Fig. 5D). We found that *Alox5* was significantly elevated in the GM:F344 tumor tissues (up 2.5-fold) compared to the GM:LEW group (Fig. 5D). Furthermore, we found that the bile acid biosynthesis genes *Cyp8b1* and *Baat* were also increased in the tumor tissues of the low tumor group (GM:LEW) compared to Pirc rats in the GM:F344 group (Fig. 5D).

We used the differential putative metabolites and differentially expressed genes in the normal epithelium to perform an integrated pathway (IP) analysis using the MetaboAnalyst module (18) to integrate metabolite, host epithelium expression, and microbiota differences. The synergistic IP analysis suggested that colonic tumor susceptibility is associated with primary bile acid biosynthesis, fatty acid elongation, and metabolism pathways. We observed increased pathway topology of unsaturated fatty acid biosynthesis corroborating the role of fatty acids in colonic tumor burden (Fig. S3B).

**TABLE 2** Genes involved in the bile acid biosynthesis and aspirin-triggered resolving E biosynthesis pathways[a]

| Pathway | Genes | Enzyme activity | FDR-adjusted P-value | Group increased in |
|---|---|---|---|---|
| Bile acid biosynthesis, neutral pathway | HSD3B7 | 3-beta-hydroxysteroid dehydrogenase type 7 | 0.861931 | NA[b] |
| | ACAA2 | 3-ketoacyl-CoA thiolase, mitochondrial | 0.321851 | NA |
| | AKR1D1 | 3-oxo-5-beta-steroid 4-dehydrogenase | 0.837208 | NA |
| | CYP8B1 | 7-alpha-hydroxycholest-4-en-3-one 12-alpha-hydroxylase | 0.00627491 | GM:LEW |
| | AMACR | Alpha-methylacyl-CoA racemase | 0.78729 | NA |
| | BAAT | Bile acid-CoA: amino acid N-acyltransferase | 0.0206231 | GM:LEW |
| | SLC27A5 | Bile acyl-CoA synthetase | 0.395957 | NA |
| | SCP2 | Chenodeoxycholoyl-CoA synthase | 0.85358 | NA |
| | CYP7A1 | Cholesterol 7-alpha-monooxygenase | 0.0779986 | NA |
| | POR | Cholesterol 7-alpha-monooxygenase | 0.093983 | NA |
| | ACOX2 | Peroxisomal acyl-coenzyme A oxidase | 0.0756656 | NA |
| | CYP27A1 | Sterol 26-hydroxylase | 0.0999591 | NA |
| | SLC27A2 | Very long-chain acyl-CoA synthetase | 1 | NA |
| | CYP2R1 | Vitamin D 25-hydroxylase | 0.853166 | NA |
| Aspirin-triggered resolving E biosynthesis | PTGS2 | 18R-hydro(peroxy)-EPE synthase | 0.000469693 | GM:F344 |
| | ALOX5 | 5S hydroperoxy HEPE synthase | 0.000469693 | GM:F344 |

[a]The genes listed in the table are part of the putative metabolite pathways differentially regulated between the high and low tumor GM groups. The predicted enzyme activity is listed adjacent to gene names.
[b]NA, not applicable.

To improve the power of our predictive analytical capacity, we used canonical correlation analyses to determine the interplay between the operational taxonomic units (OTUs), putative metabolites, and the genes identified as differential in the normal epithelium. Interestingly, we found that OTUs such as *Prevotella* sp., *Desulfovibrio* sp., and *Veillonella parvula* are associated with the GM:LEW group in the ordination plot (Fig. 6). Similarly, unannotated genes such as *Trim5* and *Tlr1*, along with *Crabp2, Junb,* and *Cndp2,* separate along the axes, based on their relationship with either GM:F344 or GM:LEW. While a putative metabolite identified as vigabatrin correlated with GM:LEW, the other metabolites detected (triethylene glycol, Intiquinatine, Ataciguat, and M659T35_2) clustered with GM:F344 in the analysis along with *Parabacteroides gordonii* (Fig. 6).

## Human colonic cancer profiles reveal mutations in the identified differential genes

Since our findings were in a preclinical rat model of colon cancer, we explored The Cancer Genome Atlas (TCGA) (19, 20) to identify whether colon cancer patients would exhibit mutations in the genes identified in our transcriptome analyses. To achieve this, we filtered the 785 colon adenocarcinoma patient data set using the differentially expressed genes involved in the bile acid pathway biosynthesis along with the *ALOX5* and *PTGS2* genes. Surprisingly, we found that 8.5% of the patients in the data set ($n = 67$; Fig. S5A) across several tumor stages (Fig. S5B) and both sexes (Fig. S5C) had a high number of mutations (>400; Fig. S5D). Similar to the phenotype observed in the Pirc rat model, 22.4% of the tumors were observed in the sigmoid colon (Fig. 5E) with a majority of those exhibiting a well-to-moderate tumor grade (Fig. 5F). Furthermore, the tumors also revealed a high microsatellite instability (MSI; 46.3%; Fig. 5G) alongside a low CpG island methylator phenotype (CIMP; 44.8%; Fig. 5H).

## DISCUSSION

Colon cancer etiology has been addressed for decades from the perspective of host gene expression and its effect on disease susceptibility. Studies have also addressed the metabolome associated with adenomagenesis separately or in conjunction with the

**TABLE 3** Putative metabolites contributing to bile acid and aspirin-triggered resolving E biosynthesis[a]

| Pathway | Putative metabolites | METLIN ID | KEGG ID |
|---|---|---|---|
| Bile acid biosynthesis, neutral pathway | (25R)−3alpha,7alpha,12alpha-trihydroxy-5beta-cholestan-26-oate | NA[b] | Null |
| | Glycocholate | 202 | C01921 |
| | Glycochenodeoxycholate | 203 | C01921 |
| | Adenosine monophosphate (AMP) | 34478 | C056466 |
| | (25R)−5beta-cholestane-3alpha,7alpha,12alpha,26-tetraol | 43029 | C00020 |
| | 7alpha,12alpha-dihydroxy-5beta-cholestan-3-one | 43117 | C05446 |
| | 7alpha,12alpha-dihydroxycholest-4-en-3-one | 43118 | C05453 |
| | 7alpha-hydroxycholest-4-en-3-one | 43126 | C17339 |
| | (25R)−3alpha,7alpha-dihydroxy-5beta-cholestan-26-al | 57924 | C05455 |
| | (25R)−3alpha,7alpha,12alpha-trihydroxy-5beta-cholestan-26-al | 57926 | C01301 |
| | (25R)−3alpha,7alpha-dihydroxy-5-beta-cholestanate | 63323 | C04554 |
| Aspirin-triggered resolving E biosynthesis | Resolvin E1 | NA | C18171 |
| | 18R-hydroxy-eicosapentaenoate | NA | Null |
| | 5S hydro(peroxy),18R-hydroxy-eicosapentaenoate | NA | Null |
| | (5Z,8Z,11Z,14Z,17Z)-icosapentaenoate | 6423 | C06428 |
| | Resolvin E2 | 36355 | C18173 |

[a]The table lists the putative metabolites involved in the bile acid biosynthesis and aspirin-triggered resolvin E pathways. The METLIN and KEGG identification numbers are also listed for testing in the future.
[b]NA, not available.

microbiome or the transcriptome (21, 22). However, the majority of these studies are retrospective (23), i.e., after disease onset in patients, raising the question of whether the microbiome, metabolome, and transcriptome are merely responding to disease, or causative of tumor development. We present for the first time the integration of three "omics" strategies to understand tumor susceptibility in a preclinical rat model of human colon cancer. Multi-omics investigations included IP analysis combining the metabolomics and transcriptomics data, identifying potential biomarkers for disease risk from fecal samples as early as 1 month of age.

Previously, we reported that differential commensal GM altered the susceptibility of isogenic Pirc rats rederived onto surrogate dams with distinct gut microbiota (9). We also showed that adenomas can develop through multiple possible mechanisms including somatic recombination and subsequent loss of the WT *Apc* allele, secondary point mutations, or through loss of expression of the WT allele (17). We report here that the GM influences the mechanism of WT *Apc* loss. Additionally, it appears that some complex GMs can possibly promote epigenetic silencing while others likely promote loss of heterozygosity.

We now report that the altered GM profile correlates with differential metabolite features representative of the high tumor (GM:F344) and the low tumor (GM:LEW) profiles. Some of the differential putative metabolites have identities in the MET-LIN database, facilitating future testing of these compounds and their influence on tumorigenesis. We remarkably found that Pirc rats with fewer adenomas (≤9, average) differentially clustered from animals with an excess of 19 adenomas, irrespective of the GM. These metabolite data were prognostic at 1 month of age, substantially prior to the onset of visible adenomas and physiological signs of disease in Pirc rats, suggesting a role for the metabolites in the initiation or early growth of adenomas. Due to the inadequacy of compound libraries in their current state, we could not establish accurate identities of the compounds using tandem MS spectra. However, further investigation including advanced methods such as UHPLC-MS-SPE-NMR (solid phase extraction and nuclear magnetic resonance) could elucidate the identity of these metabolites (24, 25). Furthermore, our analyses are limited to differential profiles and hierarchical clustering. Although, we validated the metabolomics targets within an independent cohort of PIRC and control rats, predictive analyses using naïve or unknown samples will be needed in the future. We acknowledge these caveats in our study; however, the information herein may be used going forward as training data sets for neural network or machine learning

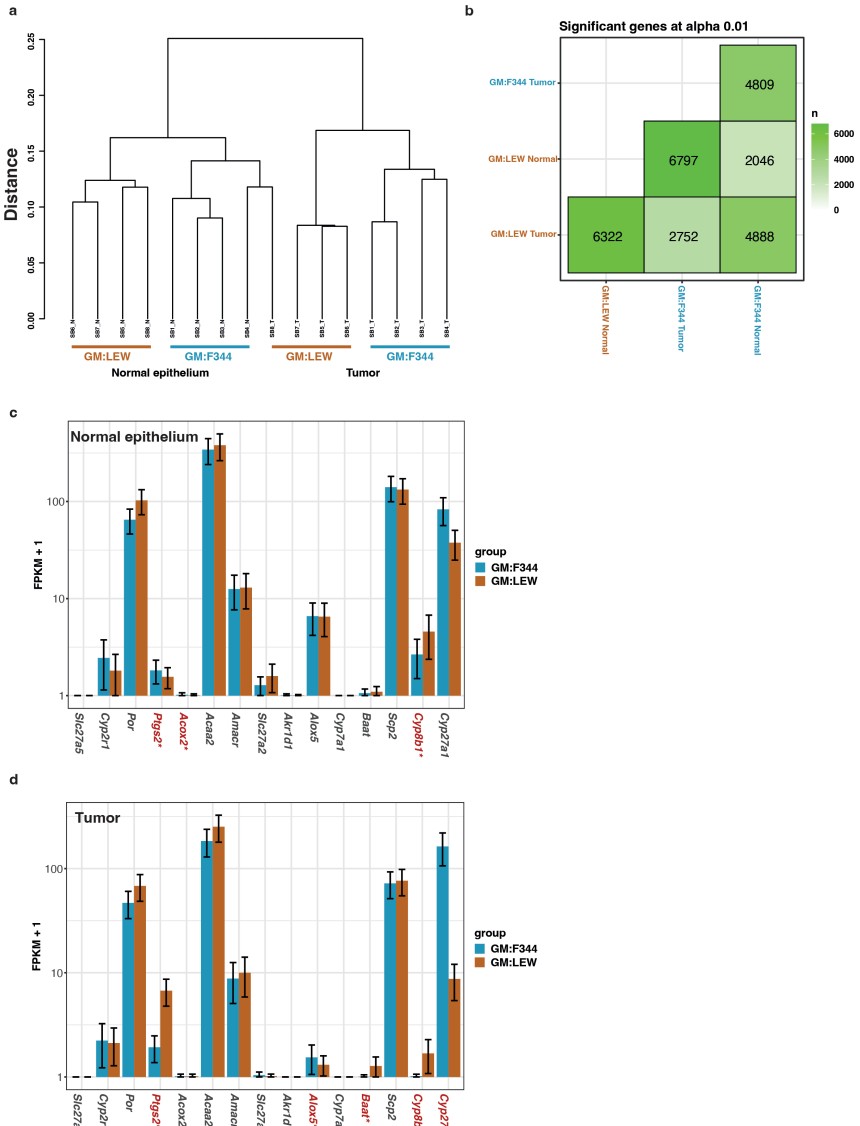

**FIG 5** GM correlates with differential gene expression in the normal epithelium and tumor tissues. Ordination and hierarchical clustering (A) analyses were used to determine the relationship of the samples to each other and the groups with respect to the other. (B) Correlation analyses between the normal colonic epithelium (normal) and tumor tissues from both GM:F344 and GM:LEW groups are indicated based on the overlap of the number of genes significantly different when Bonferroni-adjusted *P* < 0.01. Bar plots (GM:F344, cyan; GM:LEW, brown with standard deviation) depicting the relative expression of the genes involved in the pathways affected by the putative metabolites were assessed in the normal epithelium (C) and tumor (D) samples. All the analyses were performed using the cummeR-bund package in R. Relative expression of the genes involved in the pathways affected by the putative metabolites was assessed in the normal epithelium (C) and tumor (D) samples. Significantly differential expression with a Bonferroni-adjusted *P*-value less than 0.01 is identified with an asterisk (*) after the gene name.

algorithms with the objective of establishing a pre-tumorigenesis data set to identify at-risk populations based on metabolite features (26, 27).

Similar to the metabolomic analysis, we demonstrate here for the first time that co-isogenic Pirc rats show differential gene expression depending on the GM they harbor. We found that *Ptgs2* was significantly elevated in the normal epithelium in the GM:F344 group suggesting that the gut microbiota likely has a role in the differential

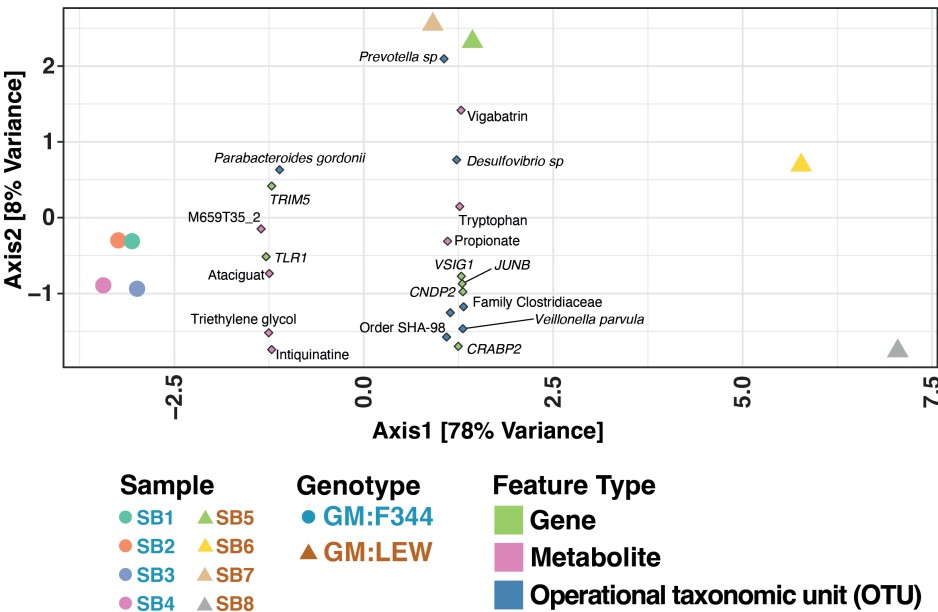

**FIG 6** Pathway and correlation analyses identify potential mechanisms and differential factors contributing to low and high tumor susceptibility. Sparse canonical correlation analysis incorporating the genes, metabolites, and OTUs contributing to disease susceptibility in GM:F344 (filled circles) and GM:LEW (filled triangles) was analyzed in R, using the structSSI CRAN package. Metabolites, genes, and OTUs are shown as diamonds in purple, green, and blue. Axis-1 demonstrated a 78% separation between GM:F344 and GM:LEW.

expression of this gene. *PTGS2* is an integral gene in the cyclooxygenase-2 (*COX2*) pathway and has been associated with increased colonic tumor burden (28). We also found increased *Alox5* expression in GM:F344 tumors, which is associated with increased proliferation and invasion of colonic tumors (29). Interestingly, the *COX2* mechanism highlighted in our study is based on metabolic pathways obtained from fecal samples collected at 1 month of age suggesting that the microbiome may be mediating low-level long-term effects through the metabolites. Furthermore, increased bile acid exposure in the gastrointestinal tract is a known factor for gastrointestinal (GI) cancers and was proposed as a carcinogen as early as 1939 (30, 31), with SBAs reported to be significantly increased in serum from patients with colonic adenomas (32). In accordance with these reports, we found that bile acid biosynthesis was elevated in the high tumor group (GM:F344). Using an IP analysis based on differential metabolites and the differentially expressed genes in the normal epithelium, we found primary bile acid biosynthesis as one of the principal contributors to the variability in disease susceptibility observed in GM:F344 or GM:LEW rats. It is imperative to note though that we found both metabolites and genes hinting toward bile acids as early as 1 month of age, prior to the onset of observable disease.

We concurrently used sparse canonical correlation analysis (sCCA) to integrate the microbiome, metabolome, and the transcriptome to identify potential features associated with disease phenotype and susceptibility. This approach was crucial to increase our confidence of prognostic feature detection as the metabolite identifications have not yet been verified through more-advanced methods such as NMR. Based on this approach, we found two genes (*Trim5* and *Tlr1*) correlated with the increased tumor phenotype of the GM:F344 group. Interestingly, these two genes (*Trim5*: tripartite motif 5 and *Tlr1*: toll-like receptor 1) are significant contributors to the innate immune system. TRIM proteins in humans are E3 ubiquitin ligases that are known to contribute to inflammatory cytokine production (33). Like the TRIM5 proteins, toll-like receptors (TLRs) activate the adaptive immune response in cancers. Lu et al. recently showed that TLR1 expression was elevated along with other TLRs in colorectal cancer (CRC) patients and

cancer cell lines. The elevated expression was associated with the simultaneous increase in expression of pro-inflammatory genes such as IL-6 (interleukin-6) which has been identified as a potential marker for diagnosis and predicting severity of CRC (34–36). The association of these genes with the high tumor group (GM:F344) suggests that the adenoma development in our model may be mediated via inflammation and/or that their expression is a response to the resident taxa. However, due to the putative identities and nomenclature of the metabolites and the genes, respectively, future studies will be required to validate their association with increased disease susceptibility. We also found that the relative abundance of OTUs (*Prevotella* and *Desulfovibrio*) (37–39), previously linked with reduced colon cancer, were simultaneously associated with the low tumor susceptibility group (GM:LEW) (9). Surprisingly, the role of *Prevotella* in CRC is not straightforward. Although, we have previously reported the association of this genera, including the species *P. copri*, with a reduced tumor phenotype, others have found it to exacerbate tumor burden. For example, Lo et al. recently reported the enrichment of *P. intermedia* in CRC, coupled with additive effects alongside *Fusobacterium nucleatum* (40). Similar other reports have indicated the association of *Prevotella* with tumor-associated mucosa in CRC individuals (41). Given this disparity within individual genera with respect to their role in CRC, our findings, especially in the context of the role *Prevotella* plays, will need to be systematically addressed. Using bacterial supplementation methods, the importance of these taxa and their role in colon cancer will also need to be carefully addressed going forward. Strikingly, we found that a majority of the differentially expressed genes in our study were found in 8.5% of patients within the TCGA data set. We used information such as genes involved in the differential metabolite pathways at an early timepoint. Interestingly, the role of SBAs such as deoxycholic acid has previously been reported to cause genomic instability leading to possible initiation and progression of CRC (42). According to Payne et al., various mechanisms by which SBAs contribute to genomic instability include oxidative DNA damage, p53 and other mutations, MSI, micronuclei formation, and aneuploidy. Concurrently, our observations highlighting the dysregulation of bile acid genes in CIMP patients are not surprising. Fennel et al. reported that bile acid metabolism genes were enriched in a particular subtype of CIMP individuals (43). They hypothesized that there may be a subset of CRC that may be arising due to an accumulation of certain bile acids, in line with previous report by Payne et al. It is intriguing that the GM:F344 group had a high prevalence of tumors that exhibit monoallelic silencing of *Apc*; however, methylation analysis of *Apc* and the genome of tumors will need to be addressed in future studies. Furthermore, an in-depth analysis of the various mutations occurring in each gene both in the PIRC rats and also in human CRC patients is necessary for a thorough understanding of disease initiation and progression. An equally important consideration in the future will be to account for other genes that do not demonstrate any mutations and how the complex interplay of various mutations influence tumor burden within CRC. Despite these challenges, these findings may be used to identify putative biomarkers for the early diagnosis of colon cancer. We recognize the limitations of this study including the sample size and the putative identities of the metabolites and recommend the validation of our findings in an independent cohort, both in preclinical models and in patients' clinical samples. While it is recognized that the GM of rat and human harbors species-specific bacteria, the metabolic capacities are similar, but strides are being made to humanize the GM in model species and could provide direct testing of bacteria that are associated with driving this differential susceptibility and mechanism of adenoma initiation (44).

Importantly, we found that the microbiome, metabolome, and transcriptome play a critical role in the etiology of colon cancer, with the GM significantly influencing the other two components. Assimilating these "omics" strategies has led to the discovery of several targets in all three systems that in the future could be used for screening and potentially therapeutic interventions. While methods for combining data from different modalities need greater development, our data signal that this approach could inform

precision medicine both in a diagnostic and prognostic manner in the future. More importantly, we demonstrated that the complex GM is an important factor that needs to be defined or controlled for in all human and animal studies examining drug or therapeutic interventions because of the altered metabolic profile and the host response.

## MATERIALS AND METHODS

### Animal husbandry and housing

Pirc rats were generated by crossing male F344/Ntac-$Apc^{+/am1137}$ Pirc rats with wild-type female rats obtained commercially from Envigo Laboratories (Indianapolis, IN, USA), i.e., F344/NHsd. All animals were group housed prior to time of breeding on ventilated racks (Thoren, Hazleton, PA, USA) in micro-isolator cages. Cages were furnished with corn cob bedding and rats were fed with irradiated 5058 PicoLab Mouse Diet 20 (LabDiet, St. Louis, MO, USA). Rats had *ad libitum* access to water purified by sulfuric acid (pH 2.5–2.8) treatment followed by autoclaving. Prior to breeding, fecal samples were collected from both the breeders using aseptic methods and banked at −80°C.

All procedures were performed according to the guidelines regulated by the Guide for the Use and Care of Laboratory Animals, the Public Health Service Policy on Humane Care and Use of Laboratory Animals, and the Guidelines for the Welfare of Animals in Experimental Neoplasia and were approved by the University of Missouri Institutional Animal Care and Use Committee.

### Experimental design

We used previously collected fecal and tissue (normal epithelium or tumor) samples from F344-$Apc^{+/am1137}$ PIRC rats generated through complex microbiota targeted rederivation as described by Ericsson et al. (9). These previously banked samples were used in this study to assess how the GM affects the metabolome and transcriptome (Fig. 1). Briefly, fecal samples collected from animals aseptically at 1 month of age and prior to onset of observable colonic tumor phenotype for metabolomics were collected and immediately snap frozen with liquid nitrogen and stored at −80°C until processing for metabolomics. At 6 months of age, animals were sacrificed post-disease onset, confirmed through colonoscopies described previously (28). Tumor (T) and adjacent normal epithelium (NE) tissues were collected into cryovials aseptically, flash-frozen, and stored at −80°C.

### Genotyping and animal identification

Pups were ear punched prior to weaning at 18 days of age using sterile technique. DNA was extracted using the "HotSHOT" genomic DNA preparation method previously outlined (45). Briefly, ear punches were collected into an alkaline lysis reagent (25 mM NaOH and 0.2 mM EDTA at a pH of 12). The ear clips were heated at 90°C on a heat block for 30 min, followed by addition of the neutralization buffer (40 mM Tris-HCl) and vortexing for 5 s. Obtained DNA was used for a high-resolution melt analysis as described previously (9).

### Serum collection

Pirc and WT rats were anesthetized with isoflurane at 1 month of age. Also, 0.5 mL of blood was drawn aseptically via the jugular vein and the serum was collected by precipitating the cells at 10,000 g for 10 min. The collected serum was centrifuged again at 16,000 for 5 min to remove any lysed debris or cells and then stored at −80°C until further processing in glass vials.

### UHPLC-MS

Fecal samples were lyophilized at −20°C using 0.1 millibar of vacuum pressure, following which dried samples (30 mg) were extracted sequentially for both UHPLC-MS and gas

chromatography-mass spectrometry (GC-MS). The dried samples were first treated with 1.0 mL of 80% MeOH containing 18 µg/mL umbelliferone, sonicated for 5 min, and centrifuged for 40 min at 3,000 g at 10°C. Also, 0.5 mL of supernatant was used for UHPLC-MS analysis after a subsequent spin at 5,000 g at 10°C for 20 min and transferring 250 µL of the sample into glass autosampler vials with inserts. For GC-MS analyses of primary polar metabolites, 0.5 mL water was added the remaining extract used above for the UHPLC preparation, sonicated for 5 min, extracted for 30 min, and centrifuged at 3,000 g. In addition, 0.5 mL of the polar extract was subsequently dried under nitrogen and derivatized using previously established protocols (46). Briefly, N-methyl-N-(trimethylsilyl) trifluoroacetamide with 1% 2,2,2-Trifluoro-N-methyl-N-(trimethylsilyl)-acetamide, chlorotrimethylsilane (TMCS) was used to derivatize the polar metabolites, after treatment with methoxyamine-HCl-pyridine. UHPLC-MS analyses were performed on a Bruker maXis Impact quadrupole-time-of-flight mass spectrometer coupled to a Waters ACQUITY UPLC system. Separation was achieved on a Waters C18 column (2.1 × 150 mm, BEH C18 column with 1.7 µm particles) using a linear gradient composed of mobile phase A (0.1% formic acid) and B (B: acetonitrile). Gradient conditions: B increased from 5% to 70% over 30 min, then to 95% over 3 min, held at 95% for 3 min, then returned to 5% for re-equilibrium. The flow rate was 0.56 mL/min and the column temperature was 60°C.

Mass spectrometry was performed in the negative electrospray ionization mode with the nebulization gas pressure at 43.5 psi, dry gas of 12 L/min, dry temperature of 250°C, and a capillary voltage of 4,000V. Mass spectral data were collected from 100 to 1,500 m/z and were auto-calibrated using sodium formate after data acquisition.

Metabolites that were significantly different between each group and that contributed to the dendrogram separating low and high tumor animals were selected for targeted tandem MS (MS/MS) analysis. MS/MS spectral data were collected using the following parameters: MS full scan: 100–1,500 m/z; 10 counts; active exclusion: three spectra, released after 0.15 min; and collision energy: dependent on mass, 35 eV at 500 Da, 50 eV at 1,000 Da, and 70 eV at 2,000 Da. Mass spectra were calibrated using sodium formate that was included as a calibration segment toward the end of the gradient separation.

## Metabolomics data processing

For UHPLC-MS data, the mass spectral data were first calibrated using sodium formate and converted into netCDF file format for processing using XCMS (47) that included peak detection, deconvolution, alignment, and integration. The signal intensities were then normalized to that of the internal standard umbelliferone (abundance of metabolite/abundance of umbelliferone × 100%) and used for statistical analysis. MS/MS spectra were searched against our custom spectral library (48) and the Bruker libraries (https://www.bruker.com/products/mass-spectrometry-and-separations/metabobase-plant-libraries/), MassBank of North America (MoNA, http://mona.fiehnlab.ucdavis.edu/), and mzCloud (https://www.mzcloud.org/) for confident or putative identifications. Multivariate statistical analysis such as principal component analyses (PCAs) and analysis of variance was performed using MetaboAnalyst (http://www.metaboanalyst.ca/) after pre-treatments of the data, i.e., normalization to sum, log transformation, and auto scaling.

## Fecal DNA extraction, 16S library preparation, and sequencing

Fecal samples were pared down to 70 mg using a sterile blade and then extracted using methods described previously (49). Amplification of the V4 hypervariable region of the 16 s rDNA and sequencing was performed at the University of Missouri DNA core facility (Columbia, MO, USA) as previously described (49).

## Normal epithelium and tumor tissue collection

All animals were humanely euthanized with $CO_2$, administration and necropsied at sacrifice. The small intestine and colon from the rats were placed on to bibulous paper and then splayed opened longitudinally by cutting through the section. Using a sterile (Feather, Tokyo, Japan) scalpel blade, normal colonic epithelium tissues were scraped from the top, middle, and distal regions of the colon. Tumors in the same locations were collected by resecting half of the total tissue. All tissues were flash-frozen in liquid nitrogen and stored at −80°C. Remaining intestinal tissues were then fixed overnight in 10% formalin, which was then replaced with 70% ethanol for long-term storage until adenoma counting was performed.

## Tumor counts and measurements

Tumor counts were estimated as previously described using a M165FC (Leica, Buffalo Grove, IL, USA) microscope at 0.73× magnification (9). Briefly, the small intestine and colonic tissues were laid flat in a large petri dish (Sycamore Life Sciences, Houston, TX, USA) and covered with 70% ethanol (ThermoFisher Scientific, Waltham, MA, USA) to prevent tissue drying. Biologic forceps (Roboz Surgical Instruments Co., Inc., Gaithersburg, MD, USA) were used to gently count polyps observable under the objective. Tissues were kept hydrated throughout the entire process. Tumor sizes were measured using the Leica Application Suite 4.2, after capturing post-fixed images as previously described (9).

## RNASeq and bioinformatics analysis

Normal epithelium and tumor tissue samples were collected upon necropsy at 180 days of age and were extracted using the Qiagen AllPrep DNA/RNA mini kit (Qiagen, Germantown, MD, USA) after pre-processing using the QIAshredder (Qiagen, Germantown, MD, USA) columns to extract total RNA. The quality of RNA was then assessed using the Experion RNA StdSens analysis kit (Bio-Rad, Hercules, CA, USA). Based on the RNA quality index (RQI), 18S and 28S peaks in the chromatogram, samples were classified into high (>9), medium (>7 or <9), or poor quality (>6). Except for one sample (normal epithelium from rat 044, i.e., 044_N), all other samples were of medium or higher RQI. Total RNA was used for poly-A selection and Illumina TruSeq paired-end library preparation following manufacturer's protocols. Seventy-five base pair paired-end reads were sequenced on the Illumina MiSeq (50) platform to an average depth of $50 \times 10^6$ reads per sample. All samples were processed at the same time and sequenced on a single lane to avoid batch effects.

Sequence read alignment was done using Tophat from the Tuxedo protocol as outlined in the original publication (51). To remove adaptors and low-quality reads, Trimmomatic v.0.32 was used with standard settings (52), and then aligned to the Rat genome (Rnor_6.0) (download from https://www.ncbi.nlm.nih.gov/assembly/GCF_000001895.5 on 24 May 2019) using Tophat2 v2.0.12 with default settings (53). The aligned reads were sorted with SAMtools v1.3, followed by HTseq v0.9.1 (54). Differential gene expression was then estimated using the DESeq2 v1.18.1 (55) in R v3.4 (56). Read count distributions in the normal epithelium and tumor tissues were found to be bimodal, with genes being identified as significant based on a false-discovery rate (FDR)-adjusted $P$-value of <0.05 and with a fold-change of at least 1.5-fold. Pathway analyses were performed on the top 100 significantly upregulated genes in either GMs, i.e., GM:F344 or GM:LEW. Pathway over-representation analyses were based on hypergeometric distribution to determine the statistical significance of a particular gene to an over-represented pathway. Topology analysis was also performed using the degree centrality method and the gene-centric IP module of Metaboanalyst v3.0 (18). For the IP analysis, the list of differentially abundant putative metabolites and differentially expressed genes between the high and low tumor normal epithelium samples were used as input. Briefly, the genes and metabolites are mapped on to KEGG pathways (57) to identify pathways that are over-represented and show increased topology, the

latter signifying the relative importance to a given pathway. Enriched pathways based on this analysis were selected using a FDR-adjusted *P*-value of <0.05. A similar analysis was performed for both the NE and T samples. We also used the sCCA (58, 59) to identify the genes, OTUs, and putative metabolites that contribute to the covariation observed in disease susceptibility. This analysis allowed for detection of entities correlating with the high and low tumor phenotypes. Briefly, we *log*-transformed the bacterial abundance, retaining only the OTUs assigned to the genus-level due to inflated zero abundance levels in the abundance tables. No transformation was employed for the putative metabolite abundance and gene count tables. Thereafter, the sCCA was performed using the procedure described by Callahan et al. (60).

## Determination of *Apc* allele ratios

To determine if the WT *Apc* allele was lost in adenomas, we used targeted pyrosequencing as previously described (17). Briefly, we PCR amplified the region of the *Apc* gene containing the Pirc mutation using the Pyromark PCR kit and performed targeted pyrosequencing using the Qiagen Q24 (Qiagen, Germantown, MD, USA). The PCR cycling profile was 94°C for 3′, followed by 50 cycles of 94°C for 15″, 57°C for 1′30″, and 72°C for 2′ with a final elongation step at 72°C for 10′. Pyrosequencing was performed according to the manufacturer's protocols using PyroMark Advanced Reagents. The ratios of the two peak heights were used to determine the Pirc to WT ratio. For determining allele expression differences, cDNA from tumor RNA was made using Invitrogen Superscript VI VILO kit per the manufacturer's instructions.

## Metabolomics analyses

Mass spectral data from each sample were converted into netCDF formatted files and processed with XCMS to generate lists of mass features and their intensities (47). An average of 499 peaks was found per sample. Peaks appearing in less than a quarter of the samples in each group were ignored. One hundred seventy-five variables were removed for threshold 25%, i.e., appearance of peaks in greater than 25% of the samples per group. Variables with missing values were replaced with a small value (0.0000001) for statistical analysis purposes. The data were then normalized to sum, transformed using log normalization, and auto-scaled to ensure maximum-possible binomial distribution. The number of samples, raw peak numbers observed, and the final peak list used for each sample processed are described in Table S1.

Statistical analyses were performed based on a threshold of 2, for the fold-change analysis, with values displayed in the log-scale to observe both the upregulated and downregulated features in a symmetrical way. Principal component analysis (PCA) was performed using the prcomp package in R using the chemometrics.R script (61). Non-metric dimensional scaling is another method for ordination and was performed using the vegan package in R (62). Hierarchical clustering analysis was performed using the Euclidean distance measure using Ward's algorithm (to minimize the sum of squares of any two clusters, potentially separating only if large differences exist between groups) and displayed as a dendrogram using the hclust function in the stat package in R. To determine the metabolites contributing to the separation and rooting of the hierarchical clusters, the samples irrespective of GM were re-classified into those with "high" or "low" tumors and a LDA was performed using the LEfSe module on a high-computing Linux platform (63) with the LDA score of $\log_{10}2$ or greater being significantly differential metabolites between the high and low tumor groups.

## Statistical analyses and figures

All other statistical analyses were performed using Sigmaplot 13.0 (64) (Systat Software, San Jose, CA, USA) and graphing for figures (except Fig. 1) was prepared through GraphPad Prism version 7 for Windows (65) (GraphPad Software, La Jolla, CA, USA). *P*-values were set to identify significance at a value less than 0.05, unless otherwise

described or indicated. Correlations were performed using the linear regression module available through GraphPad Prism v7.

## Statement of significance

Fecal metabolites, influenced by the gut microbiota, correlate with colon adenoma risk and burden in a preclinical model of familial colon cancer.

## ACKNOWLEDGMENTS

The authors wish to acknowledge Miriam Hankins, Marina McCoy, Rebecca Schehr, Aaron Ericsson, and Elizabeth Bryda for assistance with fecal collection; Nathan Bivens and the MU DNA Core for assistance with 16S rDNA and RNASeq experiments; Bill Spollen and the MU Informatics Research Core Facility for assistance with software installation for data analysis; Rat Resource and Research Center; and MU Office of Animal Resources and their staff for assistance with animal husbandry.

This research was funded by grants from the University of Missouri to Dr. James Amos-Landgraf (Startup-funding; 2012–present) and the MU Research Board grant awarded to Dr. Amos-Landgraf (2017). The MU Metabolomics Center is supported by the University of Missouri Office of Research, and the Sumner Lab is supported by NSF Awards 1340058, 1139489, and 1126719. The Sumner Lab is also supported by Bruker Daltonics, Gmbh.

Experiments were conceived and designed by S.B.B., L.W.S., and J.M.A.-L. Z.L. and S.B.B. performed the experiments with reagent and instrumental contributions from Z.L. and L.W.S. Data were analyzed by S.B.B. and Z.L. All authors contributed to writing of the paper.

The authors declare that the research was conducted in the absence of any commercial or financial relationship that could be construed as a potential conflict of interest.

## AUTHOR AFFILIATIONS

[1]University of Missouri School of Medicine, Columbia, Missouri, USA
[2]Luxembourg Centre for Systems Biomedicine, University of Luxembourg, Esch-sur-Alzette, Luxembourg
[3]Department of Biochemistry, University of Missouri, Columbia, Missouri, USA
[4]University of Missouri Metabolomics Center, Columbia, Missouri, USA
[5]University of Missouri College of Veterinary Medicine, Columbia, Missouri, USA
[6]Rat Resource and Research Center, University of Missouri, Columbia, Missouri, USA

## AUTHOR ORCIDs

Susheel Bhanu Busi http://orcid.org/0000-0001-7559-3400
Lloyd W. Sumner http://orcid.org/0000-0002-4086-663X
James M. Amos-Landgraf http://orcid.org/0000-0003-2535-7746

## FUNDING

| Funder | Grant(s) | Author(s) |
| --- | --- | --- |
| National Science Foundation (NSF) | 1340058 | Lloyd W Sumner |
| National Science Foundation (NSF) | 1139489 | Zhentian Li |

## AUTHOR CONTRIBUTIONS

Susheel Bhanu Busi, Data curation, Formal analysis, Investigation, Methodology, Software, Validation, Visualization, Writing – original draft, Writing – review and editing, Conceptualization | Zhentian Lei, Conceptualization, Data curation, Formal analysis, Software, Writing – original draft, Writing – review and editing | Lloyd W. Sumner, Formal analysis, Writing – original draft, Writing – review and editing, Funding acquisition,

Project administration, Resources | James M. Amos-Landgraf, Validation, Writing – original draft, Writing – review and editing, Funding acquisition, Resources, Supervision

## DATA AVAILABILITY

The RNAseq raw and processed data files generated and analyzed during the current study are available at the NCBI Gene Expression Omnibus (GEO) database repository under the accession ID GSE120934. The associated BioProject and SRA (sequence read archive) numbers are PRJNA495060 and SRP164571, respectively. The raw data for the metabolomics analyses are hosted through the Metabolomics Workbench on the NIH Metaboloics Data Repository under the study ID ST001075 (DataTrack ID #1539) for public access.

## ETHICS APPROVAL

The study reported here was conducted in accordance with the guidelines established by the Guide for the Use and Care of Laboratory Animals and the Publish Health Service Policy on Human Care and Use of Laboratory Animals. All studies and protocols (#6732 and #8732) were approved by the University of Missouri Institutional Animal Care and Use Committee.

## ADDITIONAL FILES

The following material is available online.

### Supplemental Material

**Fig. S1 (mSystems00151-23 S0001.pdf).** Serum metabolomics profiles and pathway analyses in Pirc and WT rats.
**Fig. S2 (mSystems00151-23 S0002.pdf).** Differentially expressed genes (DEGs) and pathways altered due to GM in the normal epithelium and tumor tissues.
**Fig. S3 (mSystems00151-23 S0003.pdf).** Multi-omic integrated analysis.
**Fig. S4 (mSystems00151-23 S0004.pdf).** Bile acid biosynthesis pathway.
**Fig. S5 (mSystems00151-23 S0005.pdf).** Genes of interest identified in The Cancer Genome Atlas (TCGA).
**Supplemental Information (mSystems00151-23 S0006.pdf).** Supplemental figure and table legends.
**Table S1 (mSystems00151-23 S0007.xlsx).** XCMS processing summary.
**Table S2 (mSystems00151-23 S0008.xlsx).** Differentially expressed genes in the normal epithelium tissues of GM:F344 and GM:LEW.
**Table S3 (mSystems00151-23 S0009.xlsx).** Differentially expressed genes in the tumor tissues of GM:F344 and GM:LEW.

### Open Peer Review

**PEER REVIEW HISTORY (review-history.pdf).** An accounting of the reviewer comments and feedback.

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
