## [Reviewer comments · mSystems]

Integrated multi-omic analyses provide insight into colon adenoma susceptibility modulation by the gut microbiota

Susheel Bhanu Busi, Zhentian Li, Lloyd Sumner, and James Amos-Landgraf

Corresponding Author(s): James Amos-Landgraf, University of Missouri

Review Timeline:

Submission Date:	February 10, 2023
Editorial Decision:	March 13, 2023
Revision Received:	May 17, 2023
Accepted:	June 6, 2023

Editor: Karoline Faust

Reviewer(s): Disclosure of reviewer identity is with reference to reviewer comments included in decision letter(s). The following individuals involved in review of your submission have agreed to reveal their identity: Guy Edmund Townsend (Reviewer #2)

Transaction Report:

DOI: <https://doi.org/10.1128/msystems.00151-23>

March 13, 2023

Prof. James Amos-Landgraf
University of Missouri
Columbia

Re: mSystems00151-23 (Integrated multi-omic analyses provide insight into colon adenoma susceptibility modulation by the gut microbiota)

Dear Prof. James Amos-Landgraf:

Thank you for submitting your manuscript to mSystems. We have completed our review and I am pleased to inform you that, in principle, we expect to accept it for publication in mSystems. However, acceptance will not be final until you have adequately addressed the reviewer comments, in particular concerning the role of different *Prevotella* strains in colorectal cancer and the over-statements pointed out by the second reviewer.

Below you will find instructions from the mSystems editorial office and comments generated during the review.

Preparing Revision Guidelines

Sincerely,

Karoline Faust

Editor, mSystems

Journals Department
Reviewer comments:

Reviewer #1 (Comments for the Author):

Susheel Bhanu Busi and colleagues used different rat models with different gut microbiomes (surrogate dams) and analysed the effect of complex gut microbiomes on metabolome and host transcriptome. They identified putative metabolite profiles that predicted future severity outcome. The transcriptomic data on the tumor epithelium identified different genes that were differentially expressed during colon cancer. By integrating the different omics datasets (metabolome, transcriptome and microbiome), the authors identified different strains to be of particular importance in driving the disease and associated them with increased fatty acid metabolism, mucin and bile acid synthesis.

This kind of study is important to better understand how the microbiome is involved in colon cancer progression. Moreover, although the findings are not entirely novel, the integration of the different omics datasets allowed the identification of different pathways and genes of interest. Here a few minor comments that the authors may address.

1. The authors comment in their results from TCGA the association of the genes of interest to MSI and CIMP. They could add a paragraph in the discussion explaining how those identified genes are related to MSI and CIMP.
2. *Prevotella*, at least some species of *Prevotella*, has also been shown to be increased and detrimental in CRC. The authors may want to clearly state in their discussion the different *Prevotella* strains and their involvement in CRC as the current statement mentions that *Prevotella* is associated with reduced colon cancer.

Reviewer #2 (Comments for the Author):

This work builds on previous findings from the same group demonstrating that colon cancer susceptibility in a preclinical Familial Adenomatous Polyposis is modulated by naturally occurring gut microbiotas and identified differences in several OTUs that were differentially abundant in microbiotas associated with higher and lower tumor burden in isogenic animals. The work presented here expands on this premise by applying multi-omics strategies to identify microbial metabolites and host transcripts that mediate this phenomenon. This manuscript represents a substantial contribution to the field by identifying 1.) putative metabolites, 2.) distinct biochemical pathways, and 3.) host transcripts that are altered in tumor susceptible animals. Furthermore, the authors present an intriguing model by which these multi-omics data sets converge around two known pathways involved in carcinogenesis, bile acid synthesis and aspirin-triggered resolvin E biosynthesis. Most importantly, the authors demonstrate that these changes are detectable prior to tumor formation, suggesting that they could serve as putative colorectal cancer biomarkers and this is buttressed by the presence of mutations in these genes in human cancers. Therefore, there is sufficient general interest in the data presented in this work to warrant publication in *mSystems*.

However, there are significant weaknesses in this manuscript that make this iteration of the manuscript unsuitable for publication. The authors consistently over-state their interpretations of their findings, which I cannot endorse as written. For example, the first results section, titled: "Metabolomic features at 1-month of age predict tumor susceptibility and severity at later developmental stages" describes a principle component analysis of untargeted metabolomics data. Although this data nicely sets up the remainder of the manuscript, no predictive analyses are performed nor blindly tested with unknown samples. Another example is the section titled "Human colonic cancer profiles reveal mutations in the identified differential genes". While likely true, the authors do not breakdown how many mutations are associated with each gene nor do they control this analysis with similar examination of genes not identified in their model system.

References are inconsistently listed or absent all together, rendering manuscript review very challenging.

In Fig. 1a, Dimensions should be explained in the legend and in Fig. 1b, the x-axis and colonic tumor multiplicity should be labeled.

Fig. 3 should add a panel(s) to demonstrate which part of each pathways are altered as it relates to the explanation in the text.

In Fig. 4a, the y-axis should be labeled, in 4c&d, panels should be labeled normal epithelium or tumor and p-values should be denoted when comparing between groups.

Fig. 5a does not aid the manuscript and should be relegated to supplementary material.

Table 2 lists genes that are not significantly different between groups.

1 **Response to the Editor and Reviewers: mSystems00151-23**

2
3 N.B.: The original remarks by the editor and reviewers are provided below in boxes
4 whereas the authors' responses are listed in black type and cited line numbers from the
5 manuscript are listed in blue and bold typeface.

6
7 **Reviewer comments:**

8
9 **Reviewer #1 (Comments for the Author):**

Susheel Bhanu Busi and colleagues used different rat models with different gut microbiomes (surrogate dams) and analysed the effect of complex gut microbiomes on metabolome and host transcriptome. They identified putative metabolite profiles that predicted future severity outcome. The transcriptomic data on the tumor epithelium identified different genes that were differentially expressed during colon cancer. By integrating the different omics datasets (metabolome, transcriptome and microbiome), the authors identified different strains to be of particular importance in driving the disease and associated them with increased fatty acid metabolism, mucin and bile acid synthesis.

This kind of study is important to better understand how the microbiome is involved in colon cancer progression. Moreover, although the findings are not entirely novel, the integration of the different omics datasets allowed the identification of different pathways and genes of interest. Here a few minor comments that the authors may address.

10
11 **Response:** *We thank the reviewer for acknowledging our study and identifying the efforts*
12 *involved in multi-omic integration. We have addressed the concerns raised by the*
13 *reviewer as indicated in our point-by-point response below.*

14
Comment 1.1. The authors comment in their results from TCGA the association of the genes of interest to MSI and CIMP. They could add a paragraph in the discussion explaining how those identified genes are related to MSI and CIMP.

15
16 **Response:** *We thank the reviewer for this comment and agree with their suggestion. In*
17 *previous work we had shown that Pirc rat adenomas can develop through either a loss of*
18 *heterozygosity pathway or through a mechanism that maintains heterozygosity but either*
19 *losses expression of the wt allele or acquires a secondary point mutation but retains*
20 *expression of both alleles. The MSI pathway has not been shown to be applicable in Pirc*
21 *tumors as this requires complete loss of mismatch repair genes. However, the reviewers*
22 *comment was so compelling we performed allele targeted pyrosequencing to determine*
23 *if adenomas develop through the LOH or MOH pathways. We found significant*
24 *differences in the percentage of tumors that developed while maintaining heterozygosity.*
25 *These adenomas showed a high rate monoallelic expression indicating that a likely*
26 *epigenetic silencing of the wt allele is likely. These data have been included in the revised*

27 manuscript as an updated **figure 1 (lines 130-145)**, the methods have been updated to
28 include the pyrosequencing methods and we have revised the discussion accordingly.
29 We have additionally included a section in the Discussion in **lines 313-318 and 408-432**,
30 expanding on how our findings may be associated with CIMP. More extensive epigenetic
31 profiling of adenomas from these two complex microbiomes would require future funding
32 and beyond the scope of this paper, but excellent suggestions by the reviewer.
33

Comment 1.2. Prevotella, at least some species of Prevotella, has also been shown to be increased and detrimental in CRC. The authors may want to clearly state in their discussion the different Prevotella strains and their involvement in CRC as the current statement mentions that Prevotella is associated with reduced colon cancer.

34
35 **Response:** We agree with the reviewer's assessment of the role of Prevotella in CRC.
36 As suggested, we have now included an extended discussion on the effect this genera
37 has in CRC individuals in the revised manuscript in **lines 394-403**.

38
39 **Reviewer #2 (Comments for the Author):**

This work builds on previous findings from the same group demonstrating that colon cancer susceptibility in a preclinical Familial Adenomatous Polyposis is modulated by naturally occurring gut microbiotas and identified differences in several OTUs that were differentially abundant in microbiotas associated with higher and lower tumor burden in isogenic animals. The work presented here expands on this premise by applying multi omics strategies to identify microbial metabolites and host transcripts that mediate this phenomenon. This manuscript represents a substantial contribution to the field by identifying 1.) putative metabolites, 2.) distinct biochemical pathways, and 3.) host transcripts that are altered in tumor susceptible animals. Furthermore, the authors present an intriguing model by which these multi omics data sets converge around two known pathways involved in carcinogenesis, bile acid synthesis and aspirin triggered resolvins biosynthesis. Most importantly, the authors demonstrate that these changes are detectable prior to tumor formation, suggesting that they could serve as putative colorectal cancer biomarkers and this is buttressed by the presence of mutations in these genes in human cancers. Therefore, there is sufficient general interest in the data presented in this work to warrant publication in mSystems.

40
41 **Response:** We thank the reviewer for highlighting and understanding our contributions
42 as being substantial to the field. We appreciate the positive feedback and also recognise
43 the concerns raised by the reviewer. We have now addressed all the comments in a
44 systematic manner below and made the necessary adjustments to the manuscript and
45 figures as suggested.
46

Comment 2.1. However, there are significant weaknesses in this manuscript that make this iteration of the manuscript unsuitable for publication. The authors consistently overstate their interpretations of their findings, which I cannot endorse as written. For

example, the first results section, titled: "Metabolomic features at 1-month of age predict tumor susceptibility and severity at later developmental stages" describes a principle component analysis of untargeted metabolomics data. Although this data nicely sets up the remainder of the manuscript, no predictive analyses are performed nor blindly tested with unknown samples. Another example is the section titled "Human colonic cancer profiles reveal mutations in the identified differential genes". While likely true, the authors do not breakdown how many mutations are associated with each gene nor do they control this analysis with similar examination of genes not identified in their model system.

47

48 **Response:** We recognise the reviewer's concerns regarding overstatements within the
49 manuscript. As highlighted, we have addressed the respective sections adjusting the
50 language for moderation. However, we would like to state that our prediction analyses is
51 based not only off of principal component analyses but also the hierarchical clustering
52 methodology as highlighted in the revised manuscript in lines 148-157, 266-278 and 332-
53 336. Furthermore, in line with the reviewer's comments, we used serum analyses in
54 control and PIRC rats to confirm our findings with respect to the metabolites being
55 enriched in animals with CRC potential. We have clarified this further in the Results and
56 also the Discussion sections in the revised manuscript in lines 332-336.

57

58 Regarding the reviewer's additional suggestions, in response to reviewer #1 in comment
59 1.2, we have already highlighted the possible role of bile acids in MSI and CIMP
60 phenotypes and must reiterate that additional mutation analyses are out of the scope of
61 this particular study. However, we acknowledge the reviewer's comment and have
62 expanded the discussion to include these suggestions and potential future directions. This
63 is now described in the revised manuscript in lines 408-439 in the revised manuscript.

64

Comment 2.2. References are inconsistently listed or absent all together, rendering manuscript review very challenging.

65

66 **Response:** We regret this oversight and have now included additional references where
67 appropriate.

68

Comment 2.3. In Fig. 1a, Dimensions should be explained in the legend and in Fig. 1b, the x-axis and colonic tumor multiplicity should be should be labeled.

69

70 **Response:** We thank the reviewer for this comment and have now updated Figure 1a to
71 represent the axes as PCoA1 and PCoA2 with the respective variation explained by them.
72 We have also adjusted Figure 1b as suggested by the reviewer.

73

Comment 2.4. Fig. 3 should add a panel(s) to demonstrate which part of each pathways are altered as it relates to the explanation in the text.

74
75
76
77
78

Response: *We thank the reviewer for this comment and have now included an additional panel indicating their alterations in the respective groups. We have also adjusted the figure legend in accordance with this suggestion.*

Comment 2.5. In Fig. 4a, the y-axis should be labeled, in 4c&d, panels should be labeled normal epithelium or tumor and p-values should be denoted when comparing between groups.

79
80
81
82
83

Response: *The figures have been adjusted as suggested by the reviewer. To avoid overcrowding of the figure, we have highlighted the p-value in the figure legend, when testing for significant difference between the groups.*

Comment 2.6. Fig. 5a does not aid the manuscript and should be relegated to supplementary material.

84
85
86
87

Response: *We have now moved Figure 5a to Supplementary Figure 3b, as suggested by the reviewer.*

Comment 2.7. Table 2 lists genes that are not significantly different between groups.

88
89
90
91
92
93

Response: *We recognize the reviewer's comment regarding non-significant genes listed in the table. We wanted to give the readers the full set of genes involved in bile acid biosynthesis and the aspirin-triggered resolvin E biosynthesis pathways, with their respective enzymes. We have further listed within the table the FDR-corrected p-value and also the groups these genes are enriched in for that particular analyses.*

June 6, 2023

Prof. James Amos-Landgraf
University of Missouri
Columbia

Re: mSystems00151-23R1 (Integrated multi-omic analyses provide insight into colon adenoma susceptibility modulation by the gut microbiota)

Dear Prof. James Amos-Landgraf:

It is my pleasure to inform you that your manuscript has been accepted, and I am forwarding it to the ASM Journals Department for publication. For your reference, ASM Journals' address is given below. Before it can be scheduled for publication, your manuscript will be checked by the mSystems production staff to make sure that all elements meet the technical requirements for publication. They will contact you if anything needs to be revised before copyediting and production can begin. Otherwise, you will be notified when your proofs are ready to be viewed. Please also make sure that metabolomics data are available, as the study number provided did not return a dataset in the Metabolomics Workbench.

If you would like to submit a potential Featured Image, please email a file and a short legend to msystems@asmusa.org. Please note that we can only consider images that (i) the authors created or own and (ii) have not been previously published. By submitting, you agree that the image can be used under the same terms as the published article. File requirements: square dimensions (4" x 4"), 300 dpi resolution, RGB colorspace, TIF file format.

We recognize that the video files can become quite large, and so to avoid quality loss ASM suggests sending the video file via <https://www.wetransfer.com/>. When you have a final version of the video and the still ready to share, please send it to mSystems staff at msystems@asmusa.org.

Sincerely,

Karoline Faust
Editor, mSystems

Journals Department
E-mail: mSystems@asmusa.org